# DA-9701 (Motilitone): A Multi-Targeting Botanical Drug for the Treatment of Functional Dyspepsia

**DOI:** 10.3390/ijms19124035

**Published:** 2018-12-13

**Authors:** Mirim Jin, Miwon Son

**Affiliations:** 1Department of Microbiology, College of Medicine, Gachon University, Incheon 21999, Korea; 2Department of Health Science and Technology, GAIHST, Gachon University, Incheon 21936, Korea; 3Research Center & Phytotherapeutics Group, Viromed, Co. Ltd., Seoul 08826, Korea; mwson@viromed.co.kr

**Keywords:** functional dyspepsia (FD), DA-9701, botanical drug, multi-targeting

## Abstract

Functional dyspepsia (FD) is the most common functional gastrointestinal disorder (FGID). FD is characterized by bothersome symptoms such as postprandial fullness, early satiety, and epigastric pain or burning sensations in the upper abdomen. The complexity and heterogeneity of FD pathophysiology, which involves multiple mechanisms, make both treatment and new drug development for FD difficult. Current medicines for FD targeting a single pathway have failed to show satisfactory efficacy and safety. On the other hand, multicomponent herbal medicines that act on multiple targets may be a promising alternative treatment for FD. DA-9701 (Motilitone), a botanical drug consisting of Corydalis Tuber and Pharbitidis Semen, has been prescribed for FD since it was launched in Korea in 2011. It has multiple mechanisms of action such as prokinetic effects, fundus relaxation, and visceral analgesia, which are mediated by dopamine D_2_ and several serotonin receptors involved in gastrointestinal (GI) functions. In clinical studies, DA-9701 has been found to be beneficial for improvement of FD symptoms and GI functions in FD patients, while showing better safety compared to that associated with conventional medicines. In this review, we provide updated information on the pharmacological effects, safety, and clinical results of DA-9701 for the treatment of FGIDs.

## 1. Introduction

### 1.1. Functional Dyspepsia: Evolution of Its Definition and Criteria

Functional gastrointestinal disorders (FGIDs) are the most common functional gastroenterologic abnormalities associated with physiological and morphological disturbances, and are often accompanied by conditions including motility dysfunction, visceral hypersensitivity, altered mucosal and immune functions, altered gut microbiota, and altered central nervous system processing [1,2,3]. In particular, functional dyspepsia (FD) has been recognized as an unexplained discomfort in the upper abdomen for over 100 years, and its symptoms include excessive fullness after eating or the inability to finish a normal-sized meal and recurrent epigastric pain [3,4]. FD is associated with not only a markedly impaired quality of life and negative impact on the work place, but also a significant economic burden [5]. The global prevalence of FD has been reported as 21%, with the range being 10% to 30% among various studies and geographies [6]. The definition and diagnostic criteria of FD have evolved within the Rome process [7]. The Rome Foundation was established in the late 1980s, at a time when there was little understanding of the pathophysiology of FGIDs. In Rome II criteria [8], FD was defined as recurrent upper abdominal pain and discomfort for at least 12 weeks during the preceding year. The Rome II classification divided patients having a wide range of dyspepsia symptoms into four groups on the basis of major symptomatic patterns: Reflux-like, ulcer-like, dysmotility-like, or nonspecific FD. However, the Rome II categorization lacked reliability, which was attributable to overlapping symptoms of various FGIDs such as irritable bowel syndrome (IBS), gastroesophageal reflux disease (GERD), postprandial distress syndrome (PDS), and epigastric pain syndrome (EPS). Therefore, the Rome III committee subdivided FD into two distinct syndromes: PDS characterized by postprandial fullness and early satiety, and EPS accompanied by epigastric pain or burning [9,10]. The Rome IV consensus published in 2016 emphasized that FD should not be considered a single disorder. It retained the subclassification of FD into PDS and EPS but strengthened the notion that these symptoms are separate entities that could overlap; it also emphasized that the symptoms should be severe enough to be bothersome (i.e., a discomfort score of at least 2 on a scale of 1 to 5 in daily life) and occur more frequently than that in the normal population. PDS and EPS were defined as “bothersome early satiety or postprandial fullness for three or more days per week in 3 months with at least a 6-month history” and “bothersome epigastric pain or epigastric burning for 1 or more days per week in the past 3 months with at least a 6-month history”, respectively [3,11]. In addition, the Rome IV consensus accepted evidence that the symptoms of GERD and IBS are part of the spectrum of FD, and the brain-gut axis is an important factor in the etiology of functional gastrointestinal (GI) disorders. It further acknowledged the possibility that pathologic lesions are involved in FD development, even if FD is a functional disorder, based on the findings that duodenal inflammation and eosinophilia are associated with FD [12,13,14]. *Helicobacter pylori*-associated dyspepsia is classified separately from true FD in the Rome IV criteria; affected patients are defined as a subset of those with FD-like symptoms and *H. pylori* infection, with symptom resolution 1 year after successful eradication of *H. pylori* [11,15].

### 1.2. Pathophysiology of FD and Current Treatments

Undoubtedly, FD is a complex and heterogeneous disorder. Accumulating data indicate that multiple factors including environment factors (food and *H. pylori* infection), biological factors (duodenal inflammation, eosinophilia, and cytokines), physiological factors (acid, gastric emptying, and gastric accommodation), and psychological factors (visceral hypersensitivity, brain pain modulating circuits, and anxiety/depression) are involved in the pathophysiology of FD [7] (Figure 1). The multiple mechanisms involved in FD make successful treatment and the development of new drugs with satisfactory efficacy and safety difficult.

Historically, FD has been considered a motility disorder dominated by disturbances in gastric physiology including gastric emptying and relaxation. Although gastric emptying was not correlated with symptoms [9,10], delayed gastric emptying was reported, with an incidence of 20 to 50% among FD patients, and the overall gastric emptying of solids was 1.5 times slower in FD patients than in healthy subjects [16]. Usually, a prokinetic is administered as a first-line treatment for PDS patients [6]. Approximately 90% of the human body’s total serotonin (5-hydroxytryptamine, 5-HT) is located in enterochromaffin cells in the gut. Intestinal movement is regulated via 5-HT receptors, including 5-HT_1_, 5-HT_2_, 5-HT_3_, 5-HT_4_, and 5-HT_7_, which are activated by serotonin as their natural ligand. Serotonin receptors are drug targets for FGIDs [17,18,19,20]. After the withdrawal of cisapride, a 5-HT_4_ agonist and 5-HT_3_ antagonist, because of its cardiovascular side effects, several serotonergic drugs including tegaserod and mosapride were launched into the market. However, the efficacy and safety of these drugs are limited [21,22]. Mosapride failed to provide any benefit [23] and tegaserod was withdrawn due to cardiovascular side effects [24].

Itopride, a prokinetic agent that works as a dual dopamine D_2_ receptor antagonist and acetylcholinesterase inhibitor, was reported to have good efficacy in global patient assessments and to improve FD symptom scores in a meta-analysis [25], but it failed to demonstrate any efficacy in a clinical trial [26]. Domperidone (a dopamine D_2_ and D_3_ receptor antagonist) and metoclopramide, (a dopamine D_1_ and D_2_ receptor antagonist) were also launched; however, their side effects, which included ventricular arrhythmia and extrapyramidal movement disorder, respectively, inhibited their long-term use [27,28].

Gastric accommodation is relaxation of the fundus, the upper part of the stomach, in response to food ingestion, which is mediated by a vasovagal reflex generated via nitrergic nerve activation. Tack et al. reported impaired gastric accommodation in 40% of FD patients [10], which was associated with early satiety [10]. Drugs that relax the gastric fundus appear to improve FD symptoms, particularly PDS. Buspirone, an antidepressant having a partial agonistic effect on 5-HT_1A_, may enhance relaxation of the gastric fundus and body, and reduce the severity of symptoms including postprandial fullness, early satiety, and upper abdominal bloating [29]. Sumatriptan, a 5-HT_1B/D_ agonist, was reported to alleviate FD symptoms by restoration of gastric accommodation with fundus-relaxing effects [30]. Recently, acotiamide, an M1/M2 muscarinic receptor blocker, was approved as a fundic relaxant for FD in Japan. This drug increases acetylcholine availability in gut nerve synapses [31], and has been reported to relax the gastric fundus and accelerate gastric emptying in humans [32,33].

Gastric hypersensitivity and/or brain-gut dysfunction has been associated with an impaired connection between gastric physiology and psychology in FD [34]. Compared to that of normal subjects, patients with FGIDs have been reported to be more sensitive to balloon distension. Many studies have confirmed intolerance to gastric distention in patients with FD [35,36]. Gastric hypersensitivity seems to be associated with postprandial epigastric pain, belching, and weight loss [37]. Approximately 34% of patients with FD had a lower threshold for the first perception of stimulus, discomfort, and pain during distention of the proximal stomach by barostat [37,38]. Visceral analgesics including alosetron, a 5-HT_3_ antagonist [39], were reported to reduce visceral hypersensitivity and to be beneficial in symptom relief compared to that of placebo in clinical studies [40]. However, side effects, including constipation and ischemic colitis, were a concern [41]. Fedotozine, an opioid κ-agonist [42,43], seemed to be a visceral analgesic; however, in clinical trials, the results of an overall physician assessment of this drug were not satisfactory [44]. For EPS patients, acid suppression is beneficial, and proton pump inhibitors (PPIs) are primarily used. H_2_ receptor antagonist therapy was reported to be superior to placebo control to reduce FD symptoms [45]. However, if first-line therapies fail, centrally acting drugs may be beneficial. Tricyclic antidepressants, including amitriptyline, appeared to provide a modest benefit for FD patients, especially for those with pain [46]. Interestingly, the tetracyclic antidepressant mirtazapine, which is an H_1_ histamine receptor blocker, 5-HT_2C_ and 5-HT_3_ antagonist, and α2 adrenergic receptor blocker, seemed to be more beneficial than placebo in a preliminary randomized controlled trial of FD patients with weight loss [47,48], although it did not alter gastric function in healthy individuals [49]. Levosulpiride, an atypical antipsychotic drug and dopamine antagonist, may be efficacious in FD treatment [50]. However, more data including those from large placebo-controlled trials are needed [51]. 

Given the lack of medications with satisfactory efficacy and safety and the requirement of multiple drugs to achieve relief from various symptoms, there remains an unmet need for new treatments. Researchers have proposed that an agent capable of modulating multiple mechanisms will be more promising than an agent highly selective for a single mechanism and that a new treatment for FD should target many, if not all, pathophysiologies [52].

For centuries, herbal preparations have been traditionally used for a variety of GI disorders based on historical experiences and oriental medicine practices, but there has been a lack of scientific evidence and qualified clinical data. Herbal medicines, which generally contain multiple herbs, may be good options for FD treatment. The pathophysiology of FD is heterogeneous and multiple components of herbal medicines may enable simultaneous targeting of the multiple pathways involved in FD, enhancing efficacy compared to that of current chemical drugs targeting a single pathway [52]. DA-9701 (Motilitone) was launched as a new drug for FD in December 2011 in Korea; it is an herbal medicine with multi-acting mechanisms for FD treatment [53]. In this review, we provide updated information related to the multi-acting pharmacological effects, safety, and clinical data of DA-9701 and propose further research on it. 

## 2. DA-9701

### 2.1. Herbal Composition

DA-9701 is a botanical drug for FD treatment in Korea, which is formulated with Pharbitidis Semen and Corydalis Tuber [53]; both constituent plants have been traditionally used for treating GI disorders [54,55,56]. Corydalis Tuber, the root of *Corydalis yanhusuo* W.T. Wang (Papaveraceae), is known to control gastric juice secretion and prevent gastric [56] and duodenal ulcers [57]. Extracts from Corydalis Tuber have been used as antispasmodic agents and analgesics [54] for abdominal pain because of its soothing and tranquilizing properties. Anti-inflammatory effects of the extract or its isolated compounds have been reported [58]. Pharbitidis Semen is the seed of *Pharbitis nil* Choisy (Convolvulaceae) and has been used as a traditional medicine for the treatment of abdominal pain [55] and as a strong purgative in Chinese medicine. The extract of Pharbitidis Semen is also used to stimulate and enhance intestinal peristalsis. Compared to that of a drug comprising a single synthetized chemical compound, a botanical drug requires more complicated quality control because of the properties of natural products. DA-9701 contains various components including corydaline (CD), dehydrocorydalin, chlorogenic acid, caffeic acid, coptisine, berberine, tetrahydroberberine (THB), palmatine, and tetrahydropalmatine (THP). For batch to batch control, CD ((13S, 13aR)-2,3,9,10-tetramethoxy-13-methyl-6,8,13,13a-tetrahydro-5H-isoquinolino[2,1-b]), the main constituent of Corydalis Tuber and CA (1S, 3R, 4R, 5R0-3-[(E)-3-(3,4-)prop-2-enoyl]oxy-1,4,5-trihydroxycyclohexane-1-carboxylic acid) from Pharbitidis Semen were selected as marker compounds for DA-9701 [59].

### 2.2. Pharmacology of DA-9701

The in vitro and in vivo pharmacological study results of DA-9701 are summarized in Table 1. 

### 2.3. Effects of DA-9701 on GI Motility

The effects of DA-9701 on gastric emptying have been evaluated in not only normal animals but also models of delayed gastric emptying induced by apomorphine, cisplatin, opioids, and clonidine. [77,78,79]. The prokinetic effects of DA-9701 were comparable to those of conventional drugs such as cisapride (10 mg/kg) in normal animals and a delayed gastric emptying model at doses of 0.3–3 mg/kg [60] and in an in vitro study of interstitial cells of Cajal, indicating pacemaker activity of DA-9701 [61]. Further, DA-9701 administration accelerated gastric emptying, as shown by a ^13^C-octanoic acid breath test with repeated measurement in normal mice [62]. In rats sutured with a strain gauge force transducer, DA-9701 administration significantly restored clonidine-induced hypomotility of the gastric antrum in pre- and postprandial periods [63]. When laparotomy, atropine, or opioids were administered to animals to examine the effects of DA-9701 on delayed GI transit, DA-9701 enhanced GI transit in mice who had been subjected to laparotomy or atropine injection [60]. Further, in guinea pigs with opioid-induced bowel dysfunction, delayed GI transit was restored and fecal output was increased. In organ bath experiments using morphine-treated ileal muscle with reduced contractility, DA-9701 significantly increased the amplitude of contraction [64]. Additionally, DA-9701 administration to animals with abdominal surgery was able to ameliorate postoperative ileus (POI) by reducing delayed GI transit and improving defecation [65]. These effects might be mediated by decreased expression of corticotrophin releasing factor in the hypothalamus of DA-9701-pretreated animals with POI [66]. Diverse stresses usually induce dyspeptic symptoms [80], probably by alteration of gastric sensory and motor functions, and acute stressors are known to be associated with delayed gastric emptying in animal models and humans [81,82]. The administration of DA-9701 improved delayed gastric emptying induced by stress, including immobilization and subsequent immersion in a water bath, which was associated with inhibition of stress-induced increases in plasma levels of adrenocorticotropic hormone (ACTH) and ghrelin [67]. DA-9701 has a high affinity for multiple receptors related to GI function. It enhances gastric emptying and GI transit via dopamine D_2_ antagonism and 5-HT_4_ agonism [83]. The affinities of DA-9701 for the D_2_ and 5-HT_4_ receptors were 0.381 and 13.2 µg/mL, respectively [53]. THB and THP, two components of DA-9701, showed dopamine D_2_ receptor antagonistic effects, and their IC_50_ values against GTPγS binding to recombinant human dopamine D_2_S receptor in Chinese hamster ovary cells were 0.622 µM (211 ng/mL) and 1.32 µM (469 ng/mL), respectively (Dong-A Pharmaceutical Co., Ltd., Seoul, South Korea). There is some controversy about the coexistence of relaxation and contraction effects on the stomach. However, this can be explained by the fact that contractile receptors are dominant in the antrum region and relaxing receptors are dominant in the fundus region of the stomach [84].

### 2.4. Effects of DA-9701 on Fundic Relaxation

Currently, the barostat is the most commonly available method for measuring fundus relaxation in human patients and canine models [37,85]. To evaluate the fundus relaxation effect, two end points can be measured: Accommodation and compliance [86,87]. The gastric accommodation response enables proximal stomach relaxation to provide space for receiving foods without an increase in gastric pressure [88]. Meal-induced gastric accommodation is thought to be the most important motor index that can be studied currently. Gastric compliance, which is tested in the fasting state, is a measure of gastric tone in the resting state and seems to be related to the pain and discomfort perception threshold [87,89]. DA-9701 significantly increased gastric accommodation in Beagle dogs, shifting the pressure-volume curve toward higher volumes that were comparable to those associated with the control dogs [60]. This finding was reproduced in the same experimental system by CD, a component of DA-9701 [68]. In DA-9701-treated dogs, meal-induced gastric volumes significantly increased and persisted, a finding comparable to that achieved with sumatriptan [69]. DA-9701 may be effective in restoring restraint stress-induced food intake inhibition in rats, and this effect appears to be related to enhanced gastric accommodation by 5-HT_1A_ agonism [70]. Further, THB isolated from DA-9701 alleviated impaired gastric compliance in the rat after stress induction and relaxed the proximal stomach via 5-TH_1A_ agonism [71]. Agonism of 5-HT_1A_ and 5-HT_1B/D_ is known to mediate relaxation of the gastric fundus through activation of a nitric oxide pathway [87,90] and to decrease the visceromotor response to noxious colorectal distention via smooth muscle relaxation [89]. The affinity of DA-9701 to 5-HT_1A_ was shown to be 6.87 μg/mL. In in vitro studies, DA-9701 inhibited 5-HT-induced contraction in feline esophageal smooth muscle cells by reducing the phosphorylation of myosin light chain kinase (MLC20) [72]. Using gastric fundus muscle strips triggered with electrical field stimulation, it was demonstrated that the effect of DA-9701 on rat gastric fundus relaxation is mainly mediated by nitrergic rather than purinergic pathways [73].

### 2.5. Effects of DA-9701 on Visceral Hypersensitivity 

Visceral hypersensitivity is one of the leading targets of FD drug development. In a neonatal rat model of colon irritation with colorectal distention (CRD), administration of DA-9701 significantly decreased mean arterial pressure in a dose-dependent manner, indicating an increase in the pain threshold with CRD-induced visceral hypersensitivity [74]. Pain signals are transmitted from the peripheral regions to the spinal cord (dorsal root ganglion) where the information is processed for transfer to the central nervous system (CNS). In rats treated with DA-9701, phosphorylation of extracellular signal-regulated kinase (p-ERK), a pain-related factor of the pain transmission pathway, significantly decreased in response to CRD [76], suggesting modulation of visceral pain. Further, THP and CD, two components of DA-9701, were reported to have antinociceptive effects on visceral and somatic nociception in animals [75]. It is known that the adrenergic nervous system plays a certain role in modulating visceral nociceptive processing. Adrenergic α_2_ agonists, such as clonidine, have been shown to reduce pain perception during gastric and colonic distention [89] and to produce post-operative analgesia in humans [91]. This is probably mediated by modulation of spinal neurotransmitters at the levels of the dorsal horn, activation of descending inhibitory pathways, and/or emotional responses to visceral stimuli [92,93]. The affinity of DA-9701 to adrenergic α_2_ receptors was shown to be 4.81 μg/mL. 

## 3. Safety of DA-9701

In a single-dose toxicity study, the LD_50_ of DA-9701 was over 2,000 mg/kg, and in a repeated-dose toxicity study for 26 weeks, the no-observed-adverse-effect level (NOAEL) was 150 mg/kg in rats. The NOAEL was 100 mg/kg in both 1-week and 13-week repeated-treatment studies in dogs. DA-9701 exhibited no genotoxicity. Due to the D_2_ antagonism of DA-9701, hyperprolactinemia was the major safety concern; however, the prolactin ED_200_ of DA-9701 was approximately 70-fold lower than that of itopride (3.78 vs. 270.1 mg/kg) in rats [53]. The pharmacokinetics and CNS distribution of THB and THP from DA-9701, both having dopamine D_2_ receptor binding affinities, were examined because D_2_ antagonists such as metoclopramide have a direct effect on the CNS by crossing the blood-brain barrier [59]. A tissue distribution study revealed that THB and THP were present at high concentrations in the stomach and small intestine compared to those in the plasma following administration of various oral doses of DA-9701, indicating considerable GI distribution. Increased concentrations of THB and THP, which cross the blood-brain barrier, were measured in the brain after repeated-dose administration of DA-9701. However, the brain concentration of DA-9701 was not expected to be sufficient to exert central dopamine D_2_ receptor antagonism following oral administration of effective doses in humans [59].

## 4. Clinical Studies of DA-9701

The therapeutic efficacy of DA-9701 has been investigated in several clinical trials according to modern guidelines and these clinical studies are summarized in Table 2. 

The first clinical trial using DA-9701 was a multicenter, double-blind, randomized, and controlled trial with concealed allocation comparing the safety and efficacy of DA-9701 and itopride hydrochloride in Korea. Four hundred and sixty-four FD patients aged >20 years who were diagnosed with FD according to the Rome II criteria were randomized to the DA-9701 (30 mg t.i.d.) or itopride (50 mg t.i.d.) group. There was a 2-week medication-free run-in phase and a 4-week treatment period. Two primary efficacy end points, including the change in composite score from baseline of the eight dyspeptic symptoms and the overall treatment effect, were analyzed. Using the Nepean Dyspepsia Index (NDI) questionnaire, the impact on patient quality of life was assessed. DA-9701 significantly improved both FD symptoms and quality of life in patients with FD. The efficacy of DA-9701 was not inferior to that of itopride. Both drugs increased the NDI score of five domains without any differences between the groups. The safety profiles of both drugs were comparable. DA-9701 was well tolerated and did not show drug-related serious adverse effects [94].

To investigate the efficacy of DA-9701 for improving FD symptoms in comparison with that of pantoprazole, a PPI, and to evaluate the additive effect of DA-9701 over PPI treatment only, a multicenter, double-blind, randomized, parallel-comparative phase IV study was conducted. Three hundred and eighty-nine patients diagnosed with FD by using the Rome III criteria were allocated to three groups: 30 mg DA-9701 t.i.d., 40 mg pantoprazole q.d., and 30 mg DA-9701 t.i.d + 40 mg pantoprazole q.d. Patients in all treatment groups reported significant improvements in FD symptoms and dyspepsia-related quality of life (*p* < 0.001). The global symptomatic improvement was 60.5% in the DA-9701 group, 65.6% in the pantoprazole group, and 63.5% in the DA-9701 + pantoprazole group, as determined using a 5-point Likert scale at week 4; there were no significant intergroup differences [95].

In another study, 81 patients with minimal change esophagitis presenting with reflux or dyspeptic symptoms were enrolled and 42 and 39 patients were randomly assigned to receive either DA-9701 30 mg or placebo t.i.d., respectively. After 4 weeks, the NDI questionnaire-Korean version (NDI-K) symptom scores [96] were significantly reduced in the treatment (*p* < 0.001) and control groups (*p* < 0.001). However, changes in the symptom scores and quality of life scores did not differ between the two groups. The reflux symptom score significantly improved in the treatment group compared to that in the placebo group among patients aged 65 years or older (*p* = 0.035). Although the NDI-K symptom scores and quality of life scores improved after 4 weeks of treatment compared to baseline values in patients with minimal change esophagitis, DA-9701 did not improve the symptom scores or quality of life scores compared with those of the placebo [96].

While PD patients have impaired gastric function, which can cause altered responses to oral dopaminergic drugs, prokinetics with dopaminergic antagonism are commonly prescribed to prevent nausea and vomiting induced by anti-parkinsonian drugs. Therefore, to evaluate the clinical utility of DA-9701 in PD patients, the effects of DA-9701 on gastric motility were evaluated in PD patients by using magnetic resonance imaging (MRI). In a non-inferiority study with domperidone, 38 participants completed the 4-week treatment protocol. DA-9701 was not inferior to domperidone in terms of changes in the 2-h gastric emptying rate (GER) before and after treatment. However, a significant increase in the 2-h GER was observed only in the DA-9701 group (*p* < 0.05) without aggravation of PD symptoms [97]. 

A clinical study was conducted to evaluate the effects of DA-9701 on gastric accommodation and emptying after a meal in healthy volunteers by using 3-D gastric volume measurement with MRI. Forty healthy subjects randomly received DA-9701 or placebo. After 5 days of treatment, the subjects underwent gastric MRI (60 min before and 15, 30, 45, 60, 90, and 120 min after a liquid test meal). Whereas DA-9701 did not significantly affect gastric accommodation in healthy volunteers, pretreatment with DA-9701 increased postprandial proximal-to-distal total gastric volume ratio compared to that with placebo. Pretreatment with DA-9701 significantly increased gastric emptying compared to pretreatment with placebo [98].

In a recent prospective study conducted in 27 patients with functional constipation diagnosed based on the Rome III criteria, DA-9701 30 mg t.i.d. for 24 days was associated with a significantly reduced colonic transit time (CTT) (*p* = 0.001) and decreased segmental CTT (*p* < 0.001). In addition, all constipation-related subjective symptoms, including spontaneous bowel movement frequency, significantly improved compared to those before treatment, without any serious adverse effects [99].

## 5. Conclusion and Future Prospects

DA-9701, a natural product for FD treatment, is composed of multiple components that act on multiple targets involved in the pathophysiology of FD. DA-9701 has affinity for dopamine, serotonin, and adrenergic receptors. Therefore, it has multiple pharmacological actions through antagonistic effects on D_2_ and agonistic effects on 5-HT_4_, 5-HT_1A_, and 5-HT_1B_, which are associated with gastric emptying, relaxation, and hypersensitivity. Although clinical studies for DA-9701 have had some limitations, beneficial effects of DA-9701 on FD symptoms and GI functions have been demonstrated in FD patients. Further, the promising results indicate that extended use of DA-9701 for PD patients and functional constipation may be possible. The effective dose of DA-9701, which is well tolerated and safe, seems to be smaller than that of conventional drugs targeting a single pathway. Currently, DA-9701 is a prescription drug for FD in Korea. To be used as a medicine in the global market, further studies involving active compounds are definitely needed for a deeper understanding of the activities of DA-9701 in FD management. For example, efficacy and detailed molecular mechanism studies for each active compound targeting different receptors, alone or in combination for additive, synergistic, or even inhibitory effects, using various FD animal models are needed. Undoubtedly, larger and sophisticatedly designed clinical trials are warranted.

## Figures and Tables

**Figure 1 ijms-19-04035-f001:**
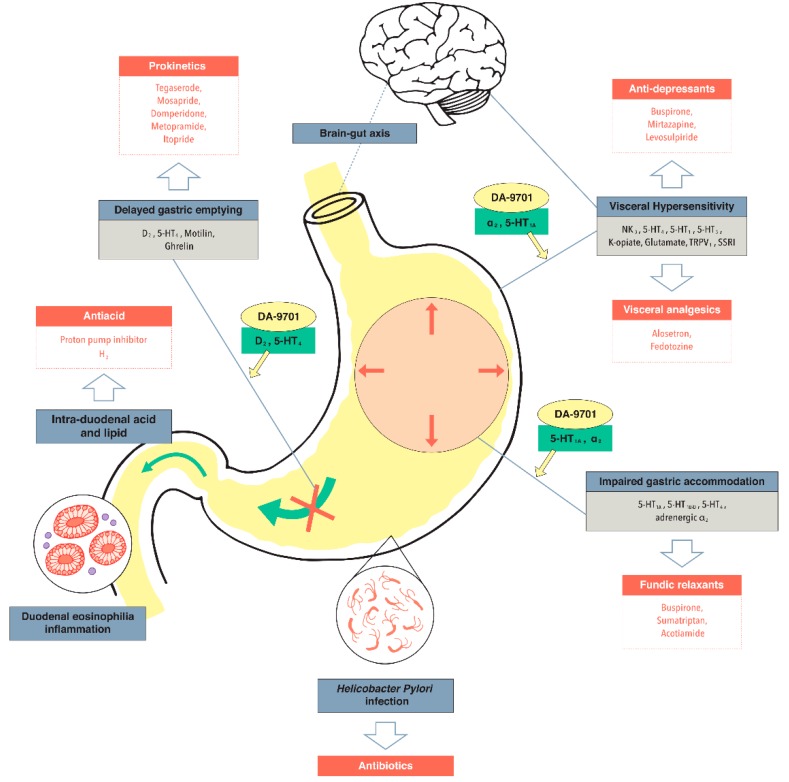
Major drug targets associated with pathophysiology of functional dyspepsia and current drugs. Targets of DA-9701 are presented.

**Table 1 ijms-19-04035-t001:** Pharmacological profile summary of DA 9701.

Class	Study Type	Experimental	Results	Ref.
Prokinetics				
D_2_ antagonistic activity	In vivo	Rat, gastric emptying delayed by apomorphine and cisplatin treatment	Acceleration of gastric emptying in normal rats, improvement in delayed gastric emptying	[60]
Gastroprokinetic activity	In vitro	Whole cell patch clamp	Modulation of pacemaker activity	[61]
Gastroprokinetic activity	In vivo	Mouse, ^13^C-octanoic acid breath test	Acceleration of gastric emptying	[62]
Gastroprokinetic activity	In vivo	Rat, clonidine-induced hypomotility of the gastric antrum	DA-9701 improved the clonidine-induced hypomotility of the gastric antrum	[63]
Gastroprokinetic activity	In vivo	Mouse, laparotomy or atropine injection	Enhancement of gastrointestinal transit	[60]
Gastroprokinetic activity	In vivo	Guinea pig, opioid-induced bowel dysfunction	Increase in amplitude of ileal muscle contraction, restoration of delayed GI transit	[64]
Gastroprokinetic activity	In vivo	Rat, postoperative ileus (POI)	Amelioration of POI by reduction in delayed GIT, improvement in defecation in rat model of POI	[65]
Prokinetic activity	In vivo	Guinea pig, postoperative ileus (POI)	Improvement in GI transit, inhibition of plasma ACTH levels via central CRF pathways	[66]
Gastroprokinetic activity	In vivo	Rat, stress-induced delayed gastric emptying	Improvement in delayed gastric emptying and inhibition of the hormonal changes induced by stress	[67]
Fundic relaxants				
Fundus-relaxing activity	In vivo	Beagle dog, canine gastric compliance with barostat	Induction of gastric relaxation and increased gastric compliance	[60]
Fundus-relaxing activity	In vivo	Beagle dog, canine gastric compliance with barostat	CD from Corydalis Tuber induced gastric relaxation and facilitated gastric accommodation	[68]
Fundus-relaxing activity	In vivo	Beagle dog, meal-induced gastric accommodation with barostat	Improvement in gastric accommodation via increase in postprandial gastric volume	[69]
5-HT_1A_ agonist activity	In vivo	Rat, restraint stress-induced feeding inhibition	DA-9701 was blocked by the 5-HT_1A_ antagonist, WAY 100635	[70]
D_2_ antagonistic activity, 5-HT_1A_ agonist activity	In vivo	Rat, restraint stress-induced impaired gastric compliance	Tetrahydroberberine, an isoquinoline alkaloid isolated from Corydalis Tuber, enhanced gastric accommodation	[71]
D_2_ antagonistic activity, 5-HT_1A_ agonist activity	In vivo	Beagle dog, canine gastric compliance with barostat	Tetrahydroberberine, an isoquinoline alkaloid isolated from Corydalis Tuber, enhanced gastric accommodation	[71]
Down-regulation of 5-HT-induced contraction	In vitro	Feline, esophageal smooth muscle cells	Inhibition of 5-HT-induced contraction via inhibition of MLC20 phosphorylation	[72]
Gastric fundus relaxation	In vivo	Rat, electrical field stimulation (EFS)-induced contractile responses	Nitrergic pathway is major mechanism involved in relaxation of rat gastric fundus by DA-9701	[73]
Visceral hypersensitivity				
Antinociceptive effect	In vivo	Neonatal rat colorectal distension (CRD)-induced visceral hypersensitivity	Significant decrease in mean arterial pressure (MAP) changes after CRD	[74]
Visceral pain modulation	In vivo	Rat, bee venom (BV)-induced persistent spontaneous nociception and pain hypersensitivity	Tetrahydropalmatine, an alkaloid constituent of plants from the genera *Stephania* and *Corydalis* effectively inhibited visceral nociception as well as thermal and mechanical inflammatory pain hypersensitivity	[75]
Visceral pain modulation	In vivo	Rat, CRD-induced visceral pain	Decrease in visceral pain via reduction in p-ERK in the dorsal root ganglion and spinal cord	[76]

**Table 2 ijms-19-04035-t002:** Clinical trials using DA-9701.

Disease	Treatment	Study Design	Sample Size	End Point	Sign	(Ref.)
FD, Rome II	DA 9701 vs. itopride	Randomized, double-blind	464 patients	Change from baseline in the composite score of the 8 dyspeptic symptoms. Overall treatment effect (OTE) as rated on a 7-grade scale.	Significant improvement in symptoms and non-inferior efficacy and comparable safety to that of itopride	[94]
FD, Rome III	DA 9701 vs. pantoprazole	Randomized, double-blind	389 patients	Global symptom assessment, response rates, difference in each score and total score of FD symptoms, difference in dyspepsia-specific quality of life (QOL) outcomes, symptomatic relief according to the subtypes of FD, i.e., epigastric pain syndrome (EPS) and postprandial distress syndrome (PDS).	Improvement in global and individual symptoms and increase in dyspepsia-specific QOL among patients. Efficacy of the monotherapy was comparable to that of pantoprazole. There was no additive effect of the combination of DA-9701 and pantoprazole.	[95]
Minimal change esophagitis	DA 9701 vs. Placebo	Double blind, placebo-controlled	81 patients	Changes in Nepean dyspepsia index questionnaire-Korean version (NDI-K) symptom score	Although NDI-K symptom scores and QOL scores improved after 4 weeks of treatment compared to baseline values in patients with minimal change esophagitis, DA-9701 did not improve the symptom scores or QOL scores compared to those of placebo.	[96]
Parkinson’s disease (PD)	DA 9701 vs. domperidone	Randomized, double-blind	40 patients	Gastric MRI, laboratory testing	Used for patients with PD to enhance gastric motility without aggravating PD symptoms	[97]
Healthy volunteers	DA 9701 vs. placebo	Randomized, double-blind, placebo-controlled	40 healthy volunteers	Gastric MRI	DA-9701 enhanced gastric emptying and did not significantly affect gastric accommodation in healthy volunteers.	[98]
Functional constipation, Rome III	DA 9701	Prospective study	37 patients	Colonic transit time (CTT) measurement	DA-9701 accelerated colonic transit and safely improved symptoms	[99]

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
