# Peer review of "DA-9701 (Motilitone): A Multi-Targeting Botanical Drug for the Treatment of Functional Dyspepsia"

_ijms, 2018, doi:10.3390/ijms19124035_

Reviewer 1 Report

General: The authors have identified an interesting research question. “Multi-targeting Botanical Drug 2 for the Treatment of Functional Dyspepsia” is an interesting topic. Functional dyspepsia is an important and emerging topic. The review methods used are appropriate and the presentation of the data is well performed. Minor corrections and grammar need to be improved before the article can be accepted.

Title is appropriate.

Abstract Looks ok

Page 3 line 97 authors mention “was reported in good rates of global assessment” setnece is not clear. Need to explain in detail.

Page 4 line 116 “studies confirmed lower colonic pain thresholds in most IBS patients and intolerance to gastric 117 distention in patients with FD” feel that bringing up ibs here is a completely different topic and is irrelevant in this section.

Page 8 line 230 “visceromoter” spelled incorrectly

Some minor spelling mistakes and grammar needs to be rechecked

This article is an updated version of the one written by the same authors in 2013

Overall a well written review.

Author Response

Responses to Reviewer 1 Comments

Manuscript ID ijms-395533

We really appreciate the reviewer's valuable comments and suggestions. We have provided our point-by-point responses below and changes are highlighted as red underlined text [A1] in the revised manuscript.

Point 1: Title is appropriate.

 Response 1: Thank you. 

Point 2: Abstract Looks ok

 Response 2: Thank you.

Point 3: Page 3 line 97 authors mention “was reported in good rates of global assessment” setnece is not clear. Need to explain in detail.

 Response 3: Thank you for bringing this to our attention. We have changed the sentence to: “Itopride, a prokinetic agent that works as a dual dopamine D2 receptor antagonist and acetylcholinesterase inhibitor, was reported to have good efficacy in global patient assessments and to improve FD symptom scores in a meta-analysis [25],” 

Point 4: Page 4 line 116 “studies confirmed lower colonic pain thresholds in most IBS patients and intolerance to gastric 117 distention in patients with FD” feel that bringing up ibs here is a completely different topic and is irrelevant in this section.

 Response 4: Thank you for your comments. We have deleted the sentence that you have indicated and have changed the sentences to: “Many studies have confirmed intolerance to gastric distention in patients with FD [35,36].”

Point 5: Page 8 line 230 “visceromoter” spelled incorrectly

 Response  5: This has been corrected to visceromotor.

Point 6: Some minor spelling mistakes and grammar needs to be rechecked

 Response  6: The minor spelling and grammar mistakes were edited by Editage.

Point 7: This article is an updated version of the one written by the same authors in 2013

 Response 7: We have updated the data over the last five years.

Point 8: Overall a well written review.

 Response  8: We really appreciate your effort to review our manuscript and your kind words.

Reviewer 2 Report

               This manuscript by Jin and Son attempt to summarize the information on the pharmacologyical effects, safety, and clinical trials of DA-9701, a botanical drug, for treatment of Functional dyspepsia (FD). The authors initiate the discussion with classification, pathophysiology, and current treatment of FD. Then they provide information about the composition and pharmacological effects of DA-9701, and then they summarize the effects of DA-9701 on GI motility, fundic relaxation, and visceral hypersensitivity. After briefly summarized the safety of DA-9701, the authors end the discussion of clinical studies of DA-9701 on treatment of FD.

               Overall, this review summarizes the information about FD and DA-9701, a botanical drug for FD treatment. Only a few minor revisions need to be done for publish.

Minor Essential Revisions

1)      Improve the English writing, there are a lot long sentences need to be re-writed, for example, the sentence from Line 16 to Line 18, and the sentence from Line 91 to Line 93.

2)      Missing several citations, such as the statement from Line 144 to Line 149, and the statement in Line 156.

Author Response

Responses to Reviewer 2 Comments

Manuscript ID ijms-395533

We really appreciate the reviewer's valuable comments and suggestions. We have provided our point-by-point responses below and changes are highlighted as red underlined text in the revised manuscript.

Point 1: Improve the English writing, there are a lot long sentences need to be re-writed, for example, the sentence from Line 16 to Line 18, and the sentence from Line 91 to Line 93.

 Response 1: The entire manuscript was thoroughly proofread by a native English speaker working for Editage. Corrections have been made, including shortening or subdividing the run-on sentences mentioned above.

Point 2: Missing several citations, such as the statement from Line 144 to Line 149, and the statement in Line 156.

 Response 2: We have now provided reference citations for the statements from lines 144 to 149 and the statement on line 156. These new references are highlighted in the reference list.
